# Knockdown of the Trehalose-6-Phosphate Synthase Gene Using RNA Interference Inhibits Synthesis of Trehalose and Increases Lethality Rate in Asian Citrus Psyllid, *Diaphorina citri* (Hemiptera: Psyllidae)

**DOI:** 10.3390/insects11090605

**Published:** 2020-09-06

**Authors:** Xinyu Liu, Zhiwen Zou, Cong Zhang, Xian Liu, Jing Wang, Tianrong Xin, Bin Xia

**Affiliations:** School of Life Sciences, Nanchang University, Nanchang 330031, China; ncuskliuxinyu@163.com (X.L.); zouzhiwen@ncu.edu.cn (Z.Z.); 352428819012@email.ncu.edu.cn (C.Z.); ct934@163.com (X.L.); wangjingyn@126.com (J.W.); xintianrong@ncu.edu.cn (T.X.)

**Keywords:** *Diaphorina citri*, trehalose-6-phosphate synthase, feeding, RNA interference, lethal phenotypes

## Abstract

**Simple Summary:**

In this study, we cloned and characterized a trehalose-6-phosphate synthase (TPS) gene from *D. citri* (*DcTPS*) for the first time. Meanwhile, we used RNA interference (RNAi) technology to efficiently disrupt *DcTPS* gene function in order to elucidate its role in the growth and development of *D. citri*. Our results suggest that dsRNA-mediated gene-specific silencing resulted in a strong reduction in relative expression of *DcTPS* and survival rate of nymphs, as well as an increase in malformation. This work was undertaken to establish a foundation for further research on the functions of *D. citri* trehalose-6-phosphate synthase. This will provide a new target for the control of *D. citri* in the field.

**Abstract:**

*Diaphorina citri* Kuwayama is the vector of citrus “huanglongbing”, a citrus disease which poses a significant threat to the global citrus industry. Trehalose-6-phosphate synthase (TPS) plays an important role in the regulation of trehalose levels of insects, while its functions in *D. citri* are unclear. In this study, full-length cDNA sequences of the TPS gene from *D. citri* (*DcTPS*) were cloned and its expression patterns at various developmental stages were investigated. The results indicated that *DcTPS* mRNA was expressed at each developmental stage and the highest *DcTPS* expression was found in the fifth-instar nymphs of *D. citri*. Additionally, mortality and deformity of *D. citri* were observed after 24 and 48 h by feeding with three different dsRNA concentrations (20, 100 and 500 ng/μL). The results indicated that *DcTPS* expression was declined, and mortality and malformation in nymphs were increased via feeding with ds*DcTPS*. Moreover, the enzyme and trehalose content were decreased, while the content of glucose was significantly higher than that of untreated (control) individuals. This suggests that *DcTPS* might be vital for the growth and development of *D. citri* and further studies of the genes should be related to molting and metabolism for controlling *D. citri*.

## 1. Introduction

The Asian citrus psyllid, *Diaphorina citri* Kuwayama (Hemiptera: Psyllidae), is a notorious pest that causes huge economic loss to the citrus industry all around the world [1,2,3]. The psyllids can excrete copious amounts of honeydew, leading to bituminous coal sickness [4]. In addition, *D. citri* serves as the only vector of the bacterium *Candidatus* Liberbacter asiaticus, which is responsible for outbreaks of “huanglongbing (HLB)”, and seriously affects economic development in the citrus industry [2,5,6,7]. Therefore, the control of *D. citri* plays a crucial role in preventing HLB from spreading [8]. Previous studies in China have shown that reduction of *D. citri* populations depended on the widespread application of insecticides [2,9,10]. Nowadays, control of *D. citri* still mainly depends on the widespread use of insecticides [11]. However, excessive use of insecticides can result in detrimental effects on human health, including poisoning, and contamination of the environment [12,13]. Therefore, exploring new and effective means of control of *D. citri* populations is urgent for sustainable economic development of the citrus industry.

RNA interference (RNAi) is a method for posttranscriptional gene silencing that has become a popular tool for exploring the function of genes in various insects [8]. Studies have shown that methods of transporting double-stranded RNA (dsRNA) into organisms mainly include microinjection, soaking, oral feeding and transgenic plant expression [14,15,16]. These methods were successfully applied to silencing of specific genes in various insects as potential pest control targets [17,18,19]. Previous research has shown that the expression of six major genes involved in the chitin biosynthesis pathway and one main gene involved in lipid catabolism was suppressed after injection with dsRNA of trehalose-6-phosphate synthase in *Heortia vitessoides* (*HvTPS*) [20]. Previous studies demonstrated that *Nilaparvata lugens* nymphs fed with ds*TPS* experienced a significant reduction in the expression of the target gene and survival rate [21]. Additionally, silencing the TPS gene of *Tribolium castaneum* resulted in malformation and an increased mortality rate; chitin metabolism of *T. castaneum* was also affected [22]. The efficacy of different dsRNA delivery methods varies with the target genes [8]. Microinjection is considered the most efficient means of knocking down target genes [23,24]. However, RNAi delivery by microinjection is difficult to achieve in small insects such as the nymphs of *D. citri* because the injection procedure can cause injury and result in abnormal death. Overall, oral feeding is considered an excellent method of gene silencing in the brown planthopper, *Nilaparvata lugens* [21]. Therefore, we hypothesized that oral feeding could be useful for silencing a gene of the nymphs of *D. citri* due to its expediency.

Trehalose, a non-reducing disaccharide formed by two glucose molecules linked by a 1α-1α bond, exists in various organisms, including plants, algae, fungus, yeasts, bacteria, insects and so on [25,26,27,28]. However, trehalose is not found in mammals [28]. Trehalose is also called “blood sugar” due to its important physiological functions in insects [29,30]. Trehalose is synthesized by trehalose-6-phosphate synthase (TPS) and trehalose-6-phosphate phosphatase (TPP) [31]. The TPS gene in insects was first reported in *Drosophila melanogaster* [32]. To date, many TPS genes have been reported from several insects, including *Heortia vitessoides* (Lepidoptera: Crambidae) [20], *Locusta migratoria manilensis* (Orthoptera: Acrididae) [33], *Tribolium castaneum* (Coleoptera: Tenebrionidae) [22], *Spodoptera exigua* (Lepidoptera: Noctuidae) [34], the brown planthopper, *Nilaparvata lugens Stål* (Hemiptera: Delphacidae) [24] and *Catantops pinguis* (Orthoptera: Catantopidae) [35]. However, the TPS mode of functionality has not been reported in *D. citri*.

In this study, *DcTPS*, a TPS gene from *D. citri,* was cloned and characterized for the first time and its expression patterns at each developmental stage were investigated by reverse transcription quantitative polymerase chain reaction (RT-qPCR). Furthermore, RNA interference (RNAi) technology was used to disrupt *DcTPS* gene function in order to elucidate its role in the development of *D. citri*, including phenotypic changes, mortality, the expression level of *DcTPS* gene, the content of trehalose and glucose and the content of trehalose-6-phosphate synthase. These findings will be conducive to further study of the role of *DcTPS* in metamorphosis and carbohydrate metabolism. Furthermore, RNAi of the *DcTPS* gene is expected to provide a technical platform and theoretical reference as a potential biological pesticide for efficient and sustainable control of *D. citri* in the future.

## 2. Materials and Methods

### 2.1. D. citri Rearing and Sample Collection

In this study, healthy adults of *D. citri* were collected from *Murraya exotica* plants in Donghu Park, Quanzhou city, Fujian Province. Then, *D. citri* was maintained in our laboratory for over 3 years and reared continuously in *Murraya exotica* in insect rearing cages (60 × 60 × 90 cm^3^). In the meantime, it was not disturbed by any insecticide. The temperature-controlled growth rooms, maintained at the laboratory building of School of Life Science in Nanchang University, were set at 27 ± 1 °C and RH (relative humidity) 70 ± 5%, with a photoperiod of 14:10 (L:D).

Following the methodology of Yu et al (2020), 100 post-mating *D. citri* females were collected and released onto fresh *Murraya exotica* plants placed in an insect rearing cage for obtaining *D. citri* individuals of the same growth and development stage. *D. citri* nymphs were classified into different stages based on their morphological features, and we continuously collected the nymphs by using a brush until the adults appeared [36]. Seven stages of *D. citri* were used for analysis of the different developmental stages’ expression levels of *DcTPS* gene including egg, first-, second-, third-, fourth-, fifth-instar nymphs and adults. Thirty individuals of *D. citri* were used for each sample collection. All stages were performed with three replicates.

### 2.2. RNA Isolation and DcTPS cDNA Synthesis

Firstly, total RNA was extracted from each sample using the Eastep^®^ Super total RNA Extraction Kit (Shanghai Promega Trading Co., Ltd., Shanghai, China). Each tube of collected sample was homogenized in an ice bath with 300 μL of lysis solution. Then, 300 μL RNA diluent was added into the tube and mixed well. The sample was heated at 70 °C for 5 min and centrifuged at 4 °C, 14,000× *g*, for 5 min. Then, 500 μL of supernatant was transferred to another new tube, and 250 μL of absolute ethyl alcohol was added and mixed well. The mixture was transferred to a new centrifugal column installed on a collecting pipe and centrifuged at 4 °C, 14,000× *g*, for 1 min. At the same time, the filtrate was discarded, and 600 μL RNA lotion was added and centrifuged at 4 °C, 14,000× *g*, for 1 min. Fifty μL of the prepared DNase 1 incubation solution was added to the adsorption film center and incubated at room temperature for 15 min. Then, 600 μL RNA lotion was added and centrifuged at 4 °C, 14,000× *g*, for 45 s, repeating the process twice. Meanwhile, the filtrate was discarded. The centrifugal column was installed on the collecting pipe and centrifuged at 4 °C, 14,000× *g*, for 2 min. Next, the centrifugal column was anew installed on an elution tube and 100 μL nuclease-free water was added. Solutions were kept at room temperature for 2 min and centrifuged at 4 °C, 14,000× *g*, for 1 min. The RNAs were stored at −80 °C. Simultaneously, the concentration and purity of RNA were assayed by NanoPhotometer N60 Touch (IMPLEN GMBH, Munich, Germany) at absorbance ratios of A260/230 and A260/280. The integrity of the total RNA was verified via 1% agarose gel electrophoresis. In accordance with the manufacturer’s instructions, total RNA was reverse-transcribed using the PrimeScript^TM^ II 1st Strand cDNA Synthesis Kit (Takara Biomedical Technology (Beijing) Co., Ltd., Beijing, China). In other words, 1.0 μL of random 6 mers, 1.0 μL of dNTP mixture and 8 μL of total RNA were mixed to reach 10 μL in the tube, which was then incubated at 65 °C for 5 min to improve reverse transcription efficiency. Then, 4.0 μL of 5×PrimeScript II Buffer, 0.5 μL of RNase Inhibitor and 1.0 μL Primer Script II RTase and RNase-free water was added to reach 20 μL. Finally, the mixture was incubated at 45 °C for 50 min and then incubated at 70 °C for 15 min. The cDNA was stored at −20 °C for subsequent experiments.

### 2.3. Molecular Cloning

Fragments of the putative *DcTPS* gene were procured from the transcriptome database for *D. citri*. The veracity of the sequences was established by polymerase chain reaction (PCR) using the primers in Table 1. Full-length cDNA was obtained by 5’- and 3’-RACE using SMARTer^®^ RACE5’/3’ kit (Takara Biomedical Technology (Beijing) Co., Ltd., Beijing, China) with the specific primers listed in Table 1. We subsequently recovered and purified the PCR product. The purified DNA was ligated onto the PGEM-Teasy Vector (Shanghai Promega Trading Co., Ltd., Shanghai, China) and the dideoxynucleotide method was used for sequencing (Sangon Biotech, Shanghai, China).

### 2.4. Bioinformatic and Phylogenetic Analyses

The cDNA sequence of *DcTPS* was translated with the Translate tool (http://www.expasy.org/translate/). Amino acid sequences were deduced using ExPASy (http://web.expasy.org/translate). The molecular weight (MW) and isoelectric point (pI) of the deduced amino acid sequences were predicted by Compute pI/Mw (http://web.expasy.org/compute_pi/). The N-linked glycosylation sites were analyzed using NetNGlyc 1.0 Server (http://www.cbs.dtu.dk/services/NetNGlyc/). Sequence comparisons were performed using DNAMAN. Additionally, a phylogenetic tree was constructed using a total of 20 insect TPS protein sequences obtained from NCBI via MEGA7.0 software and Clustal X 1.83 by the maximum likelihood method. Bootstrap values were calculated based on 1000 replicates [37].

### 2.5. Expression of DcTPS Gene

The cDNA templates derived from different developmental stages of *D. citri* were used for temporal expression tests. Primers were designed for quantitative real time (RT-qPCR) by Prime 5.0 and are listed in Table 1. Expression of the target gene was measured by RT-qPCR and normalized with two stable reference genes, β-Actin (GenBank: DQ675553) and α-tubulin gene (GenBank: DQ675550) [3]. Each PCR reaction was mixed with 10 μL TB green, 7.8 μL ddH_2_O, 1.0 μL cDNA, 0.4 μL Rox dye and 0.4 μL of each primer. The thermal cycling profile consisted of an initial denaturation at 95 °C for 5 min and 40 cycles at 95 °C for 10 s and 60 °C for 20 s. The reactions were performed with the StepOnePlus^TM^ Real-Time PCR Instrument (Thermo Fisher Scientific, Singapore). The relative expression level was calculated using the 2^−ΔΔCt^ method [38].

### 2.6. Preparation of dsRNA and Feeding

Firstly, dsRNA fragments targeting *DcTPS* (447 bp) and GFP (GenBank: LN515608) (415 bp) were prepared by using TranscriptAid T7 High Yield Transcription (Thermo Scientific, Lithuania). Secondly, the dsRNA was purified in accordance with instructions for the use of the GeneJET RNA Purification Kit (Thermo scientific, Lithuania). Thirdly, based on the results of a previous study, some adjustments were made to our rearing procedure in the dsRNA ingestion experiment [11]. Briefly, glass cylinders (12.0 cm in length and 3.0 cm in diameter) were used as a feeding chamber. The dsRNA was delivered by an artificial diet placed between two layers of stretched Parafilm. The artificial diet consisted of 20% (w:v) sucrose mixed with dsRNA *DcTPS* at a final concentration of 20, 100 and 500 ng/μL. Meanwhile, dsGFP as a control group was fed in the same way. Thirty newly emerged fifth-instar nymphs of *D. citri* were used for each treatment. Meanwhile, there were three biological replicates in each treatment. The number that survived and molted were calculated 24 and 48 h after dsRNA feeding. At the same time, individuals that were clearly of abnormal phenotype were photographed using a Leica Microsystems Ltd. (Leica, Singapore) digital camera. Subsequently, the *D. citri* individuals (at least 10 individuals) surviving different concentrations of dsRNA treatment at 24 h and 48 h were kept at −80 °C and used for RNA extraction, analysis of relative expression levels and assays of enzyme activity, trehalose content and glucose content. The sample collections of each concentration and each duration were performed with three replicates.

### 2.7. Quantitative Detection of Trehalose-6-Phosphate Synthase Content in D. citri

The content of trehalose-6-phosphate synthase in *D. citri* was measured using a modified protocol based on a previous report [20]. The content of *DcTPS* in *D. citri* was quantified by the insect trehalose-6-phosphate synthase ELISA Assay Kit (Jonln, Shanghai, China). Briefly, each sample containing three surviving individuals of *D. citri* was homogenized in 300 μL phosphate buffer saline (PBS, pH 7.0) and centrifuged at 4 °C, 5000× *g*, for 10 min. Then, 100 μL of HRP-conjugate reagent was added to 50 μL of supernatant and incubated at 37 °C for 60 min. Liquid was aspirated from each well and all wells were washed with 350 μL wash solution, repeating the process four times for a total of five washes. Then, 50 μL of chromogen solution A and 50 μL of chromogen solution B were added to each well with gentle mixing and incubated for 15 min at 37 °C under protection from light. Then, 50 μL of stop solution was added into each well. The absorbance was measured at 450 nm. The content of trehalose-6-phosphate synthase in the sample solution was calculated based on a standard curve. Three technical replicates were required for each sample measurement.

### 2.8. Measurements of Trehalose and Glucose Content

Five surviving individuals of *D. citri* were collected for the trehalose and glucose content assays. Following the assay procedure specified in the Insect Trehalose ELISA Kit and Insect Glucose ELISA Kit (Xinquan, China), samples were homogenized in an ice bath with 100 μL of extraction solution and centrifuged at 4 °C, 3000× *g*, for 10 min. Then, 40 μL of sample diluent was added to 10 μL of testing sample in each well. Then, 100 μL of HRP-conjugate reagent was added and the well was incubated at 37 °C for 1 h. Then, 50 μL of chromogen solutions A and B were mixed in after the wash followed by incubation for 15 min at 37 °C. The concentrations of trehalose and glucose were measured photometrically at 450 nm. The trehalose and glucose content in the sample solution were calculated based on a standard curve.

### 2.9. Statistical Analysis

The data were summarized as the mean ± SE (standard error) for all data sets. The data were then subjected to a one-way analysis of variance (ANOVA) using SPSS 26.0. Differences among means were tested using a Student–Newman–Keuls (S-N-K) test for multiple comparisons. All experiments were performed with three biological replicates. Each biological replicate was performed with three technical repetitions. Differences were considered statistically significant at the 5% level (*p* < 0.05).

## 3. Results

### 3.1. Sequence Analysis of DcTPS cDNA

The full-length cDNA sequence of *DcTPS* was cloned and deposited in the GenBank database (MT675285). *DcTPS* cDNA is made up of 2162 nucleotides with an open reading frame (ORF) of 1785 nucleotides (Figure 1), which encodes a protein of 594 amino acids with a predicted molecular mass of 67.057 kDa and a theoretical isoelectric point (pI) of 4.82. *Dc*TPS has two glycosylation sites. The BLAST analysis revealed that *DcTPS* shares 76% similarity identity with other insects’ TPS genes. Multiple sequence alignments showed that two signatures (HDYHL and DGMNLV) unique to TPS were well conserved in *DcTPS* (Figure 2). The phylogenetic tree showed that the *DcTPS* deduced amino acid sequence was more closely related to TPS from Hemiptera (*Acyrthosiphon pisum* and *Diuraphis noxia*) than its counterparts from Diptera, Lepidoptera, Orthoptera and Coleoptera (Figure 3).

### 3.2. Developmental Stage-Specific Expression Pattern of DcTPS

The relative expression levels of *DcTPS* at various stages were determined by RT-qPCR. The results suggested that *DcTPS* is continuously expressed at all developmental stages (F_6,14_ = 173.482, *p* = 0.0001). The expression of *DcTPS* increased steadily from the egg stage and reached a maximum in fifth-instar nymphs (Figure 4). The expression of *DcTPS* in adults declined slightly (Figure 4). As shown by the data, the expression level of *DcTPS* in the fifth-instar nymphs was 19.13 times higher than in the eggs. At the same time, the expression level in adult *D. citri* was 8.93 times higher than in the eggs. The different temporal expression patterns evident from the data suggest distinct physiological roles of *DcTPS*.

### 3.3. Phenotype and Survival Rate Analysis after Feeding with dsRNA

With the successful silencing of the *DcTPS* gene, *D. citri* subjected to RNAi exhibited abnormal phenotypes after feeding with ds*DcTPS* (Figure 5B,C). The data on survival rates of nymphs at 24 and 48 h showed that there were significant differences (*p* < 0.05) between control and 20, 100 and 500 ng/μL, with average survival rates of 97, 61, 51, 43%, respectively (Figure 6A). The decreased survival probability in *D. citri* at 24 h (F_3,8_ = 105.333, *p* = 0.0001) and 48 h (F_3,8_ = 208.889, *p* = 0.0001) can be attributed to the applied doses of ds*DcTPS* (Figure 6A).

Furthermore, after continuous feeding on the ds*DcTPS*-containing diet, the average malformation rate increased to 7, 12 and 26% on treatments after 48 h (F_3,8_ = 36.970, *p* = 0.0001) separately. This was significantly higher than malformation rates in the ds*GFP* group (Figure 6B).

To demonstrate whether target mRNA was suppressed though feeding on the ds*DcTPS* diet, the relative expression levels of the *DcTPS* gene were compared between the ds*GFP* and ds*DcTPS* (20, 100, and 500 ng/μL) treatments at 24 h (F_3,8_ = 343.308, *p* = 0.0001) and 48 h (F_3,8_ = 24.442, *p* = 0.0001). The results indicated that the relative expression level of *DcTPS* in *D. citri* decreased with increasing rates of applied ds*DcTPS* (Figure 6C). The most serious inhibition level (∼85%) was observed at 500 ng/μL after 48 h (Figure 6C).

Furthermore, compared to the ds*GFP* group (Figure 5A,D), three distinct phenotypic characteristics were evident in *D. citri* after silencing of the *DcTPS* gene. Firstly, fifth-instar nymphs were not able to complete a normal molt before death (Figure 5E). Secondly, some of the emerged adults fed with ds*DcTPS* showed “misshapen wings” (Figure 5B,F). Thirdly, partially deformed adults could not get rid of the cuticle from the nymphal stage (Figure 5C). Individuals from the ds*GFP* control group did not exhibit prominent variation in phenotype.

### 3.4. Analysis of Trehalose-6-Phosphate Synthase, Trehalose and Glucose Contents

Compared with ds*GFP*, the content of trehalose-6-phosphate synthase in *D. citri* was prominently lower at 24 h (F_3,8_ = 60.087, *p* = 0.0001) and 48 h (F_3,8_ = 66.218, *p* = 0.0001) (Figure. 7A). At the same time, the content of trehalose was also prominently lower at 24 h (F_3,8_ = 128.727, *p* = 0.0001) and 48 h (F_3,8_ = 113.991, *p* = 0.0001) (Figure 7B). However, the content of glucose was noticeably higher (Figure 7C) 24 h (F_3,8_ = 98.052, *p* = 0.0001) and 48 h (F_3,8_ = 464.652, *p* = 0.0001) after being fed with ds*DcTPS*. The results demonstrate the success of RNAi and indicate that downregulation of *DcTPS* has a considerable influence on *DcTPS* activity and the synthesis of trehalose in *D. citri*.

## 4. Discussion

In this study, a cDNA sequence encoding TPS from *D. citri* was cloned and characterized for the first time. Sequence analysis showed that there are HDYHL and DGMNLV motifs in the putative *DcTPS* amino acid sequence. This result is consistent with previous studies [20,34]. Furthermore, multiple sequence alignment and the phylogenetic tree demonstrate higher identity and a closer evolutionary relationship of *DcTPS* with TPS from other Hemipteran insects such as *A. pisum*, *D. noxia* and *N. lugens*. These results are in accordance with the fact that TPS genes of similar species might have a closer evolutionary relationship and therefore could be clustered together [3,21,34].

The expression of the *DcTPS* gene in multiple developmental stages has been reported in *S. exigua* [34], *N. lugens* [21], *L. decemlineata* [39], *H. armigera* [40] and *M. domestica* [41]. The result indicates that it might have a unique role in insect growth and development [20]. Like other insects, the expression level of the *DcTPS* gene was found to be stage specific. Its expression was highest before molting, which has been verified in many insect species [20,23,41]. Therefore, *DcTPS* may be related to molting, and this is possibly related to the requirement for trehalose during metamorphosis. Trehalose is a stored energy source for flight and can be hydrolyzed by cleavage of the glycosidic linkage to release two molecules of glucose and release of substantial energy to meet demand [26,42]. As a trehalose synthase, energy requirements of adults for flight should result in higher expression of *DcTPS* at this stage [41]. On the other hand, higher expression in adults may be also related to energy requirements for mating and spawning [26].

Many previous research reports have demonstrated the feasibility of RNAi through feeding [8,11,21,30]. Our results have shown that feeding-based RNAi of the *DcTPS* gene can specifically restrain the expression of the *DcTPS* gene and significantly affects nymphal growth and development in *D. citri*, leading to increased mortality and deformity. Consistent with these studies, RNAi of *BmTPS* and *HvTPS* decreased the larval survival rate and caused mortality or deformed phenotypes in *B. minax* [20] and *H. vitessoides* [23]. Many adults of *D. citri* did not molt normally and presented with misshapen wings after molting as a result of feeding on ds*DcTPS*. There might be many explanations for the molting abnormalities. Firstly, the regulation of chitin biosynthesis is related to the concentration of trehalose substrate and the expression of main genes in the pathway of chitin biosynthesis [21]. As a result of limited energy supply, chitin synthesis is affected when insects undergo metamorphosis [20,24,30,43]. Secondly, paucity of trehalose would weaken protection against abiotic stresses [23,29,32]. Thirdly, insect metamorphosis and wing formation are regulated by the TPS gene [21]. Meanwhile, lack of trehalose might weaken energy metabolism in healthy individuals of *D. citri* [21,41]. Therefore, there are some negative effects on the growth of nymphs and adults which cause even higher mortality and malformation in *D. citri* following the knockdown of the *DcTPS* gene. These results suggest that RNAi technology could be used for in-field control of *D. citri* [8].

Knockdown of the *DcTPS* gene significantly decreased *DcTPS* and trehalose content in *D. citri*. TPS is involved in trehalose synthesis as an important enzyme [26]. Silencing of the *DcTPS* gene could further weaken trehalose synthesis and lessen the trehalose content in *D. citri*. Consequently, a sharp drop in trehalose content could make nymphs more susceptible to stress conditions [41]. Trehalose is not only an energy store but also plays a crucial role in combating the negative effects of stress [26]. Moreover, the content of glucose increased in *D. citri* after RNAi. The reason behind this result may be that less reactants are involved in the trehalose synthesis chain so that glucose is continually accumulated. Our results are also in keeping with previous research showing that there is a negative correlation between trehalose and glucose content in insects [34].

## 5. Conclusions

Full-length cDNA of the *DcTPS* gene in *D. citri* was cloned for the first time. Developmental stage expression analysis showed that *DcTPS* expression was the highest at the fifth-instar nymph stage. In addition, there is a sharp drop in the expression of *DcTPS* and the number of surviving *D. citri* individuals attributed to dsRNA-mediated gene-specific silencing. There is also clear evidence of enhanced malformation. The significant change in trehalose and glucose concentration after RNAi suggests that *DcTPS* is related to trehalose metabolism of *D. citri.* These results establish a foundation for future studies on the physiological function of the *DcTPS* gene and provide a potential pest control target for management of *D. citri*. In our next study, we will design some novel biological insecticides which are directly targeted at *D. citri* trehalose metabolism and combined with RNAi to control *D. citri* in the field.

## Figures and Tables

**Figure 1 insects-11-00605-f001:**
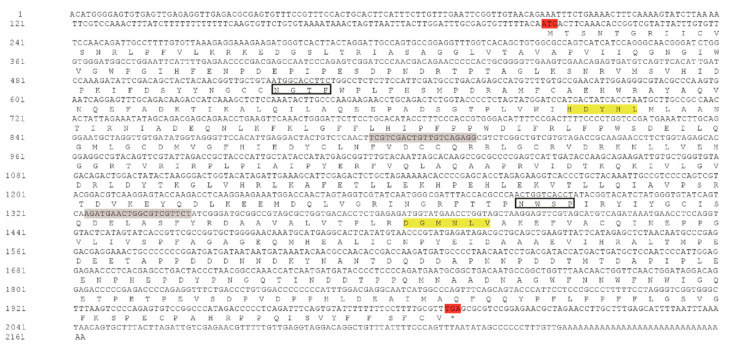
Nucleotide and deduced amino acid sequences of *DcTPS* from *D. citri*. Both initiation codon and termination codon are shaded in red. Motifs (or signature motifs) unique to trehalose-6-phosphate synthase (TPS) (residues 450–464 and 555–567) are shaded in yellow. Two glycosylation sites are indicated in black box. The primer sequences used for synthesis of dsRNA are shaded in gray.

**Figure 2 insects-11-00605-f002:**
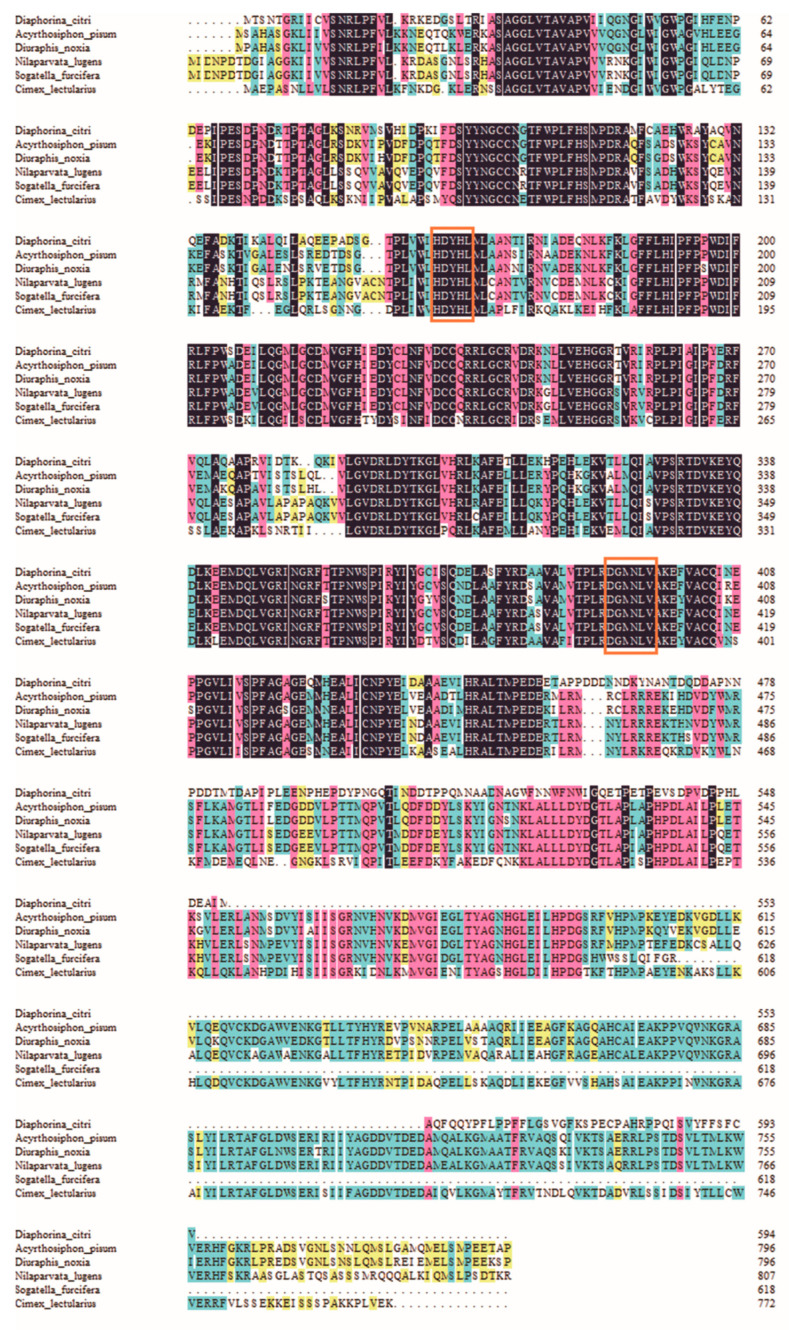
The amino acid alignment of trehalose-6-phosphate synthase (TPS) sequences from *Diaphorina citri* (MT675285), *Acyrthosiphon pisum* (XP_001944221), *Diuraphis noxia* (XP_015365486), *Nilaparvata lugens* (ACV20871), *Sogatella furcifera* (JQ013797), *Cimex lectularius* (XP_014255923). Signature motifs unique to trehalose-6-phosphate synthase (TPS) (HDYHL and DGMNLV) are indicated with red frame.

**Figure 3 insects-11-00605-f003:**
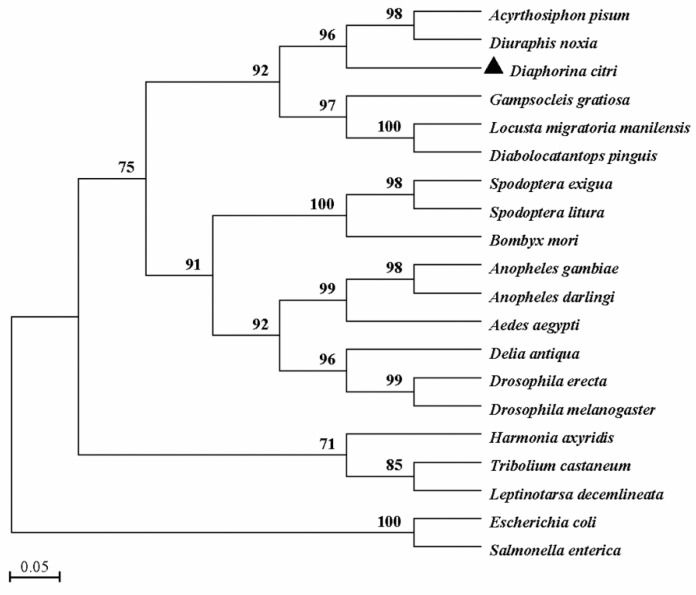
Phylogenetic analysis of *DcTPS* and TPSs from other species. The phylogenetic tree was constructed based on the amino acid sequences of known TPSs. Full-length amino acid sequences were aligned with the Mega 7.0 program to generate the phylogenetic tree by the maximum likelihood method. A bootstrap analysis was carried out, and the robustness of each cluster was verified with 1000 replicates. Values at the cluster branches indicate the results of the bootstrap analysis. The sequences were obtained from GenBank under the following accession numbers: *Acyrthosiphon pisum* (XP_001944221), *Diuraphis noxia* (XP_015365486), *Gampsocleis gratiosa* (APZ77037), *Locusta migratoria manilensis* (EU131894), *Diabolocatantops pinguis* (ACV32626), *Spodoptera exigua* (ABM66814), *Spodoptera litura* (ADA63844), *Bombyx mori* (XP_004926812), *Anopheles gambiae* (XP_317243), *Anopheles darling* (ETN66003), *Aedes aegypti* (XP_001657813), *Delia antiqua* (AFW99833), *Drosophila erecta* (XP_001968664), *Drosophila melanogaster* (NP608827), *Harmonia axyridis* (FJ501960), *Tribolium castaneum* (XP_975776), *Leptinotarsa decemlineata* (AO799586), *salmonella enterica* (PXW07297) and *Escherichia coli* (NP416410). Scale bar represents 0.05 substitutions per site.

**Figure 4 insects-11-00605-f004:**
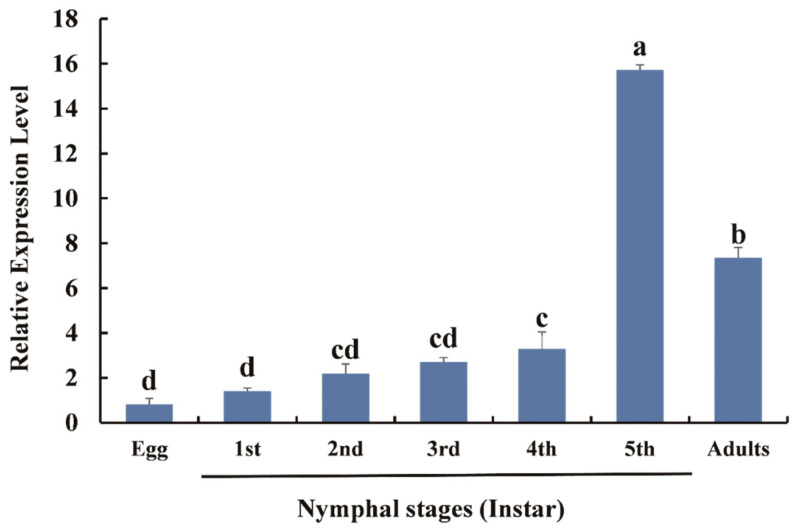
Expression patterns of *DcTPS* at different developmental stages of *D. citri*. Relative expression levels of *DcTPS* were analyzed using RT-qPCR and calculated with the 2^−ΔΔCt^ method. SPSS 26.0 software was used for statistical analysis. Bars with difference letters are significantly different according to Student–Newman–Keuls (S-N-K) test (*p* < 0.05).

**Figure 5 insects-11-00605-f005:**
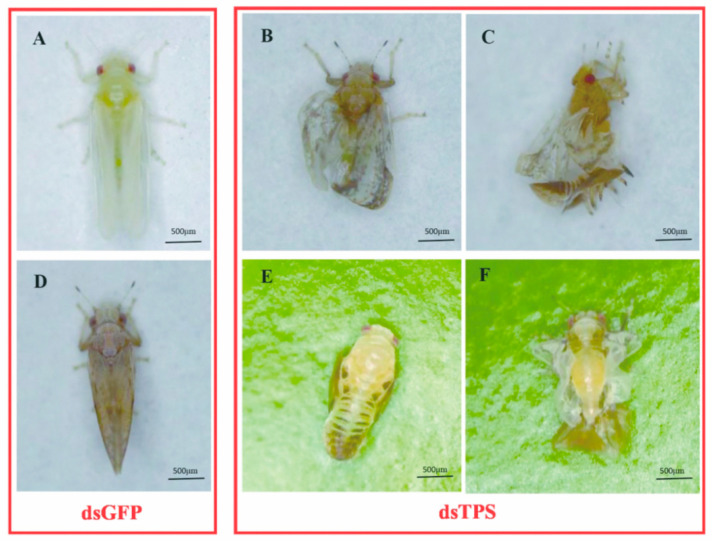
Effects of RNAi on *D. citri* representative phenotypes after feeding with dsRNA. (**A**,**D**) A newly emerged adult and mature adult come from the fifth-instar nymph treated with ds*GFP* (control). (**B**,**C**,**E**,**F**) *D. citri* that underwent metamorphosis after being treated with ds*DcTPS*.

**Figure 6 insects-11-00605-f006:**
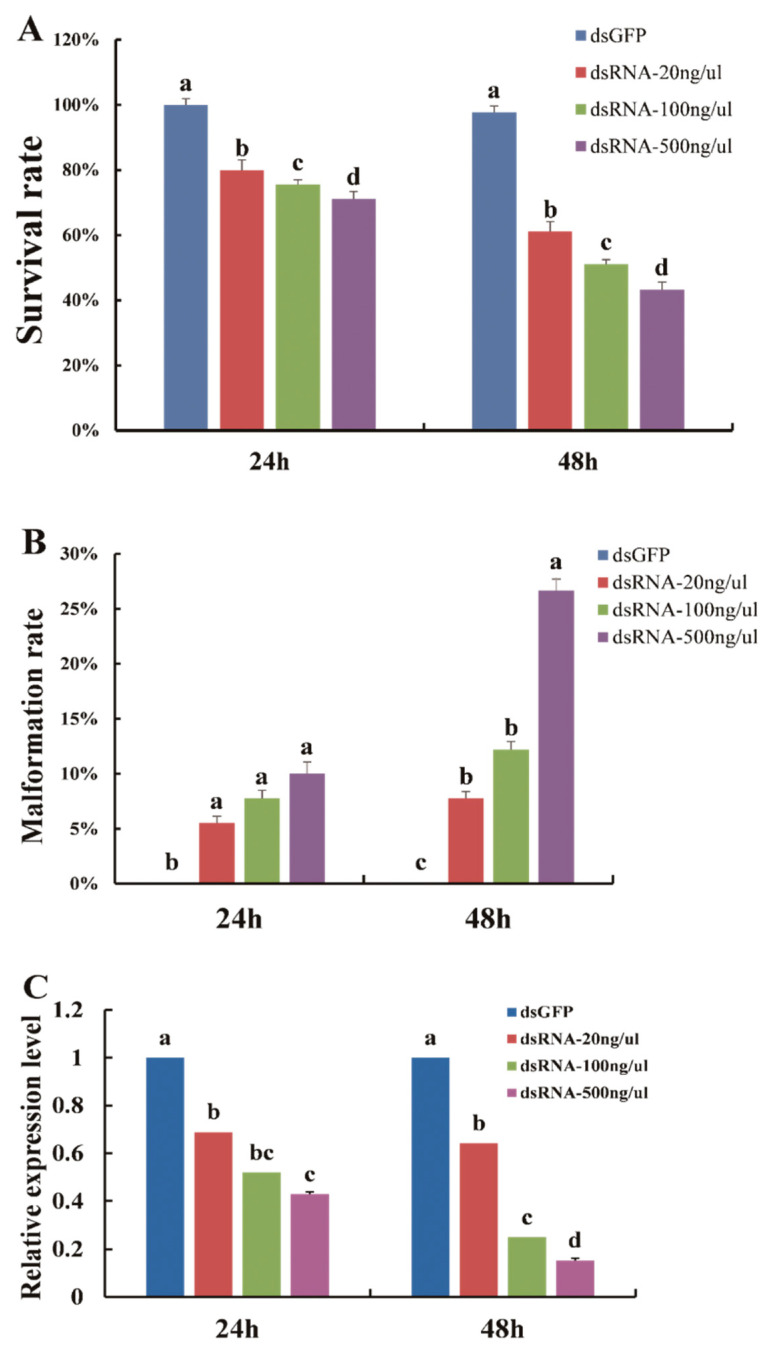
Effects of feeding ds*DcTPS* on survival rate (**A**), malformation rate (**B**) and relative expression levels (**C**) in *D. citri*. Results of control (ds*GFP*) and treatment groups (ds*DcTPS* of 20, 100 and 500 ng/μL) 24 and 48 h post RNAi are shown. Statistical analysis was conducted using SPSS 26.0 software. Bars with different letters are significantly different in terms of SNK test (*p* < 0.05).

**Figure 7 insects-11-00605-f007:**
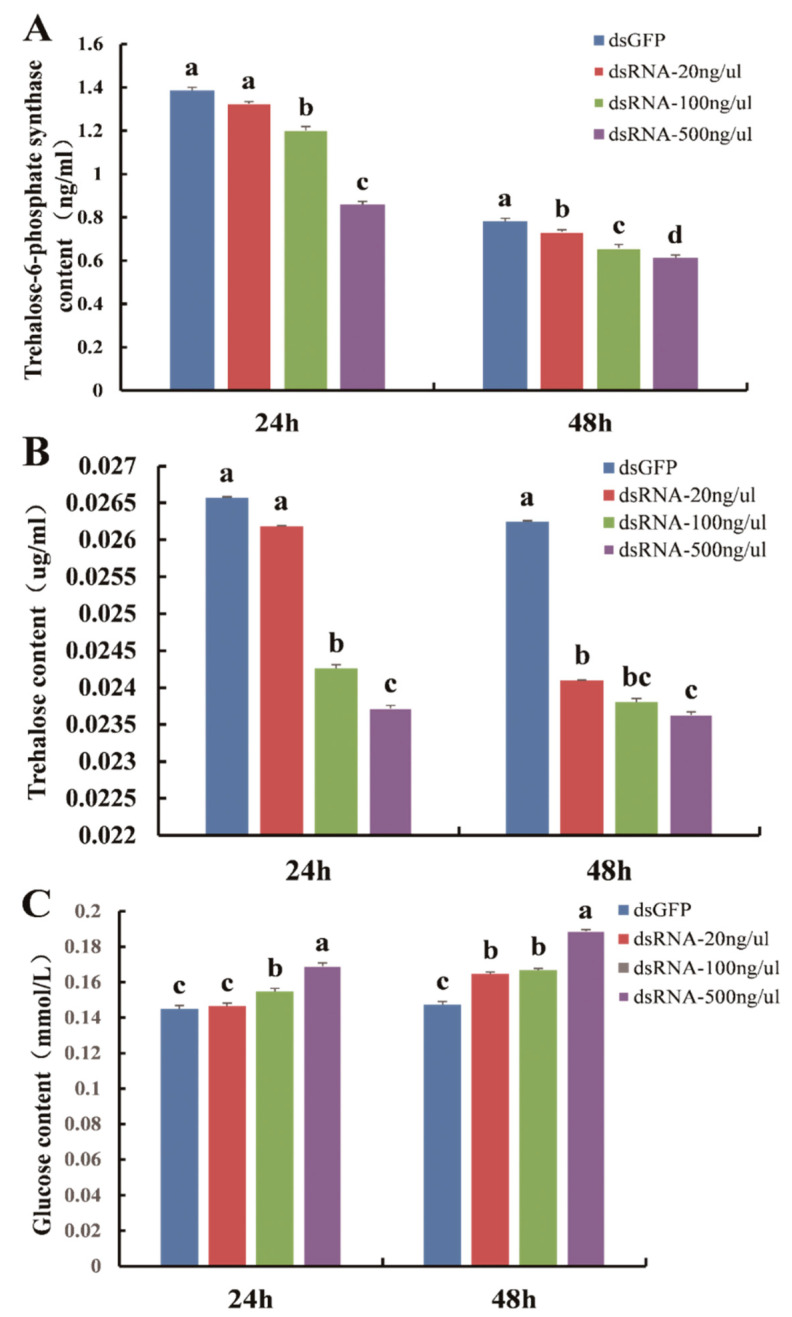
Detection of trehalose-6-phosphate synthase content (**A**), trehalose content (**B**) and glucose content (**C**) in *D. citri* after feeding dsRNA for 24 and 48 h. Bars with different letters are significantly different according to SNK test (*p* < 0.05).

**Table 1 insects-11-00605-t001:** Primers used in this study.

Primer Names	Primer Sequences	Length of Primer	Primer Usage
*DcTPS*-F	ATGCTTGCCGCCAACACT	18 bp	Middle fragment
*DcTPS*-R	CGCATCCCGATAGAACGA	18 bp	
Dc5’TPS-R1	CCTCCAATGTTCGGCACAAA	20 bp	5’RACE
Dc5’TPS-R2	CCAGAAGGTGCCATTACAGC	20 bp	
Dc3’TPS-F1	GCTCTAACAATGCCCGAGGAC	21 bp	3’RACE
Dc3’TPS-F2	ACAATGCCCGAGGACGAG	18 bp	
UPM long	CTAATACGACTCACTATAGGGCAAGCAGTGGTATCAACGCAGAGT	45 bp	
UPM short	CTAATACGACTCACTATAGGGC	22 bp	
NUP	AAGCAGTGGTAACAACGCAGAGT	23 bp	
ß-Actin-F	CCCTGGACTTTGAACAGGAA	20 bp	RT-qPCR
ß-Actin-R	CTCGTGGATACCGCAAGATT	20 bp	
α-tubulin-F	GGTTCAAGGTGGGTATCAACTAT	23 bp	
α-tubulin-R	TAGCGGTGGTGTTGGAAAG	19 bp	
Q- *DcTPS* -F	AGGGAATGCTAGGTTGTGAT	20 bp	
Q*-DcTPS* -R	TGCTCTACCAGGAGGTTCTT	20 bp	
ds*GFP*-F	TAATACGACTCACTATAGGGAAGGGCGAGGAGCTGTTCACCG	42 bp	dsRNA synthesis
ds*GFP*-R	TAATACGACTCACTATAGGGCAGCAGGACCATGTGATCGCGC	42 bp	
ds*DcTPS*-F	TAATACGACTCACTATAGGGTCGTCGACTGTTGTCAGAGG	40 bp	
ds*DcTPS*-R	TAATACGACTCACTATAGGGAGAACGACGCCAGTTCATCT	40 bp	

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
