# Peer review of "Knockdown of the Trehalose-6-Phosphate Synthase Gene Using RNA Interference Inhibits Synthesis of Trehalose and Increases Lethality Rate in Asian Citrus Psyllid, Diaphorina citri (Hemiptera: Psyllidae)"

_insects, 2020, doi:10.3390/insects11090605_

Round 1
Reviewer 1 Report
Reviewer’s comments:
This paper deals with the knockdown of the trehalose-6-phosphate synthases genes using RNA inferences inhibits synthesis of trehalose and increase lethality rate in Asian citrus psyllid, Diaphorina citri. Although the experiment methods are appropriate, the conclusion are not supported by the data presented. This paper is not suitable for published for several reasons: The method of statistical analysis is inappropriate, or statistical analysis is not performed in most data. Some of the data presented are inaccurate. For example, assays of enzyme activity data. Sample of collected ACP, RNA isolation and feeding bioassay sample size was not clearly. The genes identified data was not supported by RNA transcriptome data.
Comments:
Line 13: Please change the citrus cancer to citrus diseases.
Line 21: add “,and” between 100 and 500 ng/µL
Line 66: Please write complete common name, authority name, family and order of the pest when writing fort first time in the introduction.
Line 81: Please add the information where, when, what host was collected for laboratory population. Does it have any application insecticide for three years?
Line 85: How many ACP was released and how many days was collected sample. Please provides detailed information.
Line 93: How many ACP for extracted RNA for each stage? Please provide gel electrophoresis data for supplementary material.
Line 139: How may replication for each feeding bioassay and each replication how many instars?
Line 157 to 16: Please provide calculated method for enzyme activity.
Line 196: Provide detailed information for phylogenetic tree for supplementary data.
Line 22 to 258: Results part provides statistical values (F values, df and p values) to support your results. This is will give clear picture of significance or no significance, how many samples were used for experiment.
Author Response
Response 1: In this study, the method of statistical analysis referred to some previous studies, and this study was supported by many references on insects' trehalose-6-phosphate synthase gene RNAi. Therefore, we think our method of statistical analysis is appropriate. In addition, we have made some modifications and additions according to your suggestions, especially in quantitative detection of trehalose-6-phosphate synthase content in D. citri, sample of collected ACP, RNA isolation and so on. We hope these details of supplementary can make the article more clearly. Besides, the transcriptome data what we used has been compared with the NCBI database, and its correctness has been verified by PCR technology.
Comments:
Line 13: Please change the citrus cancer to citrus diseases.
Response: We have changed the citrus cancer to citrus diseases according to your suggestion. Thanks for your advice.
Line 21: add “,and” between 100 and 500 ng/µL
Response: We have added “and” between 100 and 500 ng/µL according to your suggestion. Thank you.
Line 66: Please write complete common name, authority name, family and order of the pest when writing fort first time in the introduction.
Response: We have added family and order name of the pest according to your suggestion. Thanks.
Line 81: Please add the information where, when, what host was collected for laboratory population. Does it have any application insecticide for three years?
Response: We have added the sampling information of D. citri. In another word, healthy adults of D. citri in our study were collected from Murraya exotica plants in Donghu Park, Quanzhou city, Fujian Province.
Line 85: How many ACP was released and how many days was collected sample. Please provides detailed information.
Response: We selected 100 individuals of post-mating D. citri females and released them to fresh Murraya exotica plants to lay eggs. D. citri nymphs were classified into different stages based on their morphological features, and continuously collected the nymphs by using a brush until the adults appear.
Line 93: How many ACP for extracted RNA for each stage? Please provide gel electrophoresis data for supplementary material.
Response: There are 30 individuals of D. citri used for extracted RNA in each treatment. At the same time, there are three individuals of each stage. This information was shown in Line 94 in our new submission manuscripts. Furthermore, we provide the agarose gel electrophoresis of total RNA from D. citri in supplementary figure 1.
Line 139: How many replication for each feeding bioassay and each replication how many instars?
Response: There are thirty newly emerged fifth-instar nymphs of D. citri used for each treatment group, and there are three biological replicates in each treatment group. These details were shown in Line 160 in our new submission manuscripts.
Line 157 to 16: Please provide calculated method for enzyme activity.
Response: It was measured by the insect trehalose-6-phosphate synthase ELISA Assay Kit (Jonln, Shanghai, China). The absorbance was measured at 450 nm. The content of trehalose-6-phosphate synthase in the sample solution were calculated based on a standard curve (y = 0.2492x + 0.0493). We have made quantitative detection of trehalose-6-phosphate synthase content in D. citri. We can use that as a parameter for enzyme activity. The details are shown in lines 171-181.
Line 196: Provide detailed information for phylogenetic tree for supplementary data.
Response: We have shown what you mentioned in the diagram in Figure 3, including the GenBank of TPSs gene from other’s species and the specific method for constructing phylogenetic trees.
Line 22 to 258: Results part provides statistical values (F values, df and p values) to support your results. This is will give clear picture of significance or no significance, how many samples were used for experiment.
Response: We have added the relevant information about the F values, df and p values in our new submission manuscripts.
Besides, I want to explain about the language of our manuscript. We have consulted professionals to help us revise this manuscript, and we can provide you with the language editorial certificate. Thank you very much.

Reviewer 2 Report
The manuscript entitled “Knockdown of the trehalose-6-phosphate synthase gene using RNA interference inhibits synthesis of trehalose and increase lethality rate in Asian Citrus Psyllid, Diaphorina citri (Hemiptera: Psyllidae)” examines cDNA cloning of trehalose-6-phosphate synthase (DcTPS) gene and determination of the gene expression patterns during growth of D. citri. Moreover, knockdown of the DcTPS gene using RNA interference (RNAi) by feeding diet containing dsRNA lead to decreases in the expression of DcTPS gene. Synthesis of trehalose and lethality rate in D. citri were increased by the RNAi. These results suggest that the DcTPS gene is required for trehalose synthesis and normal growth in D. citri.
The study provides important insight of the function of the trehalose-6-phosphate synthase in D. citri. That finding may contribute to develop new methods to inhibit proliferate D. citri for citrus industry. Further, the RNAi by feeding diet containing dsRNA was useful for analysis of function of genes in D. citri.
COMMENT:
1). Is RNA interference by feeding diet containing dsRNA in D. citri new point? In introduction, the authors described that “Furthermore, RNAi of the DcTPS gene is expected to provide a technical platform and theoretical reference as a potential biological pesticide for efficient and sustainable control of D. citri in the future.” (Page 2, lines 76-78) On the other hand, the authors described that “oral feeding is considered an excellent method for silencing TPS gene of the nymphs of D. citri due to its expediency and low casualty rate [21]”. (Page 2, lines 58-59) In the reference [21], expression of a gene of D. citri was inhibited by the RNA interference by feeding diet containing dsRNA? The technique is highly useful for study of D. citri. If the technique is novel point of this study, the authors should emphasize and carefully described introduction.
If the RNA interference by feeding diet containing dsRNA in D. citri is anew point, I recommend that the authors should change the sentence “oral feeding is considered an excellent method for silencing TPS gene of the nymphs of D. citri due to its expediency and low casualty rate [21]” to “oral feeding is considered an excellent method in the brown planthopper, Nilaparvata lugens [21]. Therefore, we hypothesized that the oral feeding could be useful for silencing a gene of the nymphs of D. citri due to its expediency.”.
Author Response
Response 2: Previous studies shown that RNA interference has been successfully applied to silencing of specific genes in D. citri by feeding diet containing dsRNA. For example, RNA interference of trehalase by feeding the artificial diet containing dsRNA affects the trehalose and chitin metabolism pathways in Diaphorina citri (Hemiptera: Psyllidae) [11]. In addition, silencing the D. citri chitin synthase (DcCHS) gene by RNAi significantly increased mortality, and dsRNA delivery was performed using an artificial die [8]. These results demonstrated the effectiveness of RNAi by feeding. Therefore, we believe that there was no effect on RNA interference by feeding diet containing dsRNA in D. citri. Furthermore, D. citri is a pest with a piercing-sucking mouthpart. Feeding is more suitable for controlling of D. citri in field than the microinjection. Because we aim to study the specific function of the DcTPS gene for controlling of D. citri in field. So we think the research on DcTPS gene and gene products are more important than dsRNA-mediated technique. Thank you very much.

Round 2
Reviewer 1 Report
Comments to Author
The present study reports the “DcTPS might be vital for growth and development of Asian Citrus Psyllid, Diaphorina citri”. I have accepted this paper to publish in the insects. However, I have a few comments.
Please check result of statistical analyses be carefully. There were a lot of wrong.
Author Response
Response: we have checked the results of statistical analyses and made minor modifications in some detail in our manuscript. In addition, minor adjustments were also made to figure 4, figure 6B, figure 7A and figure 7C to ensure the standardization of statistical results.Thanks for your suggestion.
